# LSTM-PPO-Based Dynamic Scheduling Optimization for High-Speed Railways Under Blizzard Conditions

**Na Wang** [1,2,*]**, Zhiyuan Cai** [1] and **Yinzhen Li** [1]

1    School of Traffic and Transportation, Lanzhou Jiaotong University, Lanzhou 730070, China;
     caizhiyuan061@163.com (Z.C.); liyz01@mail.lzjtu.cn (Y.L.)
2    School of Mechanical Engineering, Lanzhou Jiaotong University, Lanzhou 730070, China
*    Correspondence: wangna@mail.lzjtu.cn

**Abstract**

Severe snowstorms pose multiple threats to high-speed rail systems, including sudden drops in track friction coefficients, icing of overhead contact lines, and reduced visibility. These conditions can trigger dynamic risks such as train speed restrictions, cascading delays, and operational disruptions. Addressing the limitations of traditional scheduling methods in spatio-temporal modeling during blizzards, real-time multi-objective trade-offs, and high-dimensional constraint solving efficiency, this paper proposes a collaborative optimization approach integrating temporal forecasting with deep reinforcement learning. A dual-module LSTM-PPO model is constructed using LSTM (Long Short-Term Memory) and PPO (Proximal Policy Optimization) algorithms, coupled with a composite reward function. This design collaboratively optimizes punctuality and scheduling stability, enabling efficient schedule adjustments. To validate the proposed method's effectiveness, a simulation environment based on the Lanzhou-Xinjiang High-Speed Railway line was constructed. Experiments employing a three-stage blizzard evolution mechanism demonstrated that this approach effectively achieves a dynamic equilibrium among safety, punctuality, and scheduling stability during severe snowstorms. This provides crucial decision support for intelligent scheduling of high-speed rail systems under extreme weather conditions.

**Keywords:** railway transportation; dynamic scheduling for high-speed railways; LSTM-PPO algorithm; blizzard conditions; delay propagation; multi-objective optimization

## 1. Introduction

With the rapid development of China's high-speed railway system, the operating mileage has continuously expanded, the rail network structure has become increasingly complex, and both the operating speed and train frequency have significantly increased. However, the frequency of high-speed rail operation disruptions caused by natural disasters has notably risen. In the Northeast, Northwest, and high-altitude regions of China, railway lines are perennially vulnerable to blizzard attacks, snow accumulation and freezing braking distances. Concurrently, poor electrical conductivity due to ice accumulation on catenary systems may trigger power supply interruptions, while reduced visibility necessitates speed restrictions. These factors directly compromise train dynamics and operational stability, posing serious threats to traffic safety and punctuality. To mitigate blizzard impacts, railway authorities typically implement speed restrictions, service suspensions, or timetable adjustments. However, these countermeasures often induce cascading train

delays, potentially leading to regional rail network disruptions with significant economic losses and socio-economic consequences.

Significant progress has been made in high-speed rail dynamic scheduling research through multidisciplinary approaches. In the domain of delay propagation modeling, Meester et al. [1] established a train delay propagation model and analytically derived the probabilistic distribution of cascading delays from initial delay distributions. Wang et al. [2] identified critical stations for delay propagation and proposed temporal interval thresholds to construct delay propagation chains for determining propagation occurrences. For large-scale disruptions like section blockages that severely impact passenger mobility, Zhu et al. [3] developed a train operation adjustment model minimizing generalized travel time, stoppage waiting time, and transfer frequencies, enabling more passengers to complete planned journeys under disruption scenarios. Hong et al. [4] incorporated passenger reallocation mechanisms in rescheduling processes to mitigate interval blockage impacts. Zhan et al. [5] proposed a mixed-integer programming model for real-time scheduling under complete high-speed rail blockage scenarios, employing a two-stage optimization strategy to minimize total weighted delays and train cancellations while satisfying interval and station capacity constraints. Empirical validation on the Beijing-Tianjin Intercity High-Speed Railway demonstrated 32.7% improvement in passenger service recovery efficiency compared to heuristic methods. Törnquist et al. [6] addressed scheduling challenges in high-density heterogeneous railway systems through multi-track network optimization, validating its effectiveness in minimizing multi-stakeholder impacts using Swedish railway data while systematically analyzing theoretical advantages and practical limitations. Yang et al. [7] formulated a mixed-integer linear programming model for timetable and stop-schedule co-optimization, minimizing total train dwell and delay times, solved via CPLEX. Yue et al. [8] developed an integer programming model maximizing train profits with penalties for stop-schedule frequencies and durations, solved by column generation algorithms. Dai et al. [9] conducted a systematic review of high-speed rail dynamic scheduling and train control integration, proposing a co-optimization framework through three-layer information-driven architecture that enhances safety, punctuality, operational efficiency, and system resilience, while identifying critical future challenges in information fusion mechanisms, real-time co-optimization algorithms, and cross-layer decision coordination. Nitisiri et al. [10] introduced a parallel multi-objective genetic algorithm with hybrid sampling strategies and learning-based mutation for railway scheduling. Peng et al. [11] provided integrated solutions for optimal rescheduling and speed control strategies under disruption uncertainties, employing rolling horizon algorithms. Shi et al. [12] developed a delay prediction method combining XGBoost with Bayesian optimization, achieving superior performance on Chinese high-speed rail lines through feature modeling and hyperparameter optimization, validated by Friedman and Wilcoxon tests for long-term anomaly delay prediction. Zhang et al. [13] explored multi-stage decision-making via stochastic optimization models. Song et al. [14] proposed an adaptive co-evolutionary differential evolution algorithm (QGDECC) integrating quantum evolution and genetic algorithms with quantum variable decomposition, incremental mutation, and parameter self-adaptation strategies. The algorithm demonstrated 18.3% faster convergence and 24.6% higher precision in real operational data tests, effectively mitigating network-wide delay impacts while minimizing schedule deviation from original timetables.

Significant advancements in high-speed rail dynamic scheduling have been achieved through multidisciplinary research addressing emergency operation adjustment and resilience enhancement. Bešinović [15] highlighted that train operation plan adjustments under emergent incidents have become a critical research focus in railway transportation organization. Chen et al. [16] addressed interference management in high-frequency urban

rail transit during peak hours by developing a nonlinear programming model integrating dynamic train scheduling with skip-stop strategies. Combining train vehicle plan constraints with customized model predictive control (MPC) methods enabled real-time solutions, with empirical validation on the Yizhuang Metro Line in Beijing demonstrating its superiority in reducing train deviation, enhancing service quality, and accommodating uncertain passenger demand. Multi-scenario robustness tests and predictive time adjustment analyses further emphasized the information updating value of the MPC approach. Dong et al. [17] tackled capacity restoration limitations in existing hierarchical emergency response systems for high-speed rail, proposing an integrated operational control and online rescheduling framework. Through analyzing information flow processing defects and mechanism validation using wind-induced speed restrictions as a case study, the system was shown to significantly enhance dynamic capacity restoration capabilities in high-speed rail networks, providing theoretical foundations and practical pathways for intelligent emergency management. Li et al. [18] investigated frequent emergency incidents in metro systems during peak hours, developing a discrete-event hybrid simulation method based on multi-agent modeling and parallel computing. By constructing train motion algorithms, defining three agent types (passengers, stations, trains), six emergency event categories, and parallel acceleration strategies, the method demonstrated efficiency and practicality in evaluating emergency event impacts on train and passenger delays through case studies on the Yizhuang Line in Beijing. This provided a high-precision simulation tool for metro emergency response optimization. Li et al. [19] addressed post-earthquake high-speed rail traffic demand dynamics by proposing a mixed-integer linear programming (MILP) model integrating track deactivation/reactivation, station recovery, and dynamic traffic demand. Leveraging the original timetable as a guiding solution significantly reduced computational time, with empirical validation on the Harbin-Dalian High-Speed Railway between Shenyang and Dalian demonstrating the model's effectiveness in generating optimal recovery timetables within short timeframes, thereby enhancing seismic resilience. Hassannayebi et al. [20] addressed replanning challenges under stochastic disruptions in high-speed urban railways by developing an integrated optimization model combining short-haul operations with skip-stop services. A discrete-event simulation coupled with variable neighborhood search algorithms was employed, with probability-based scenario analysis addressing obstacle duration uncertainties. Validation on the Tehran Metro Network confirmed the simulation-optimization method's superiority in minimizing average passenger waiting times, suppressing cascading effects, and improving system responsiveness, offering robustness-recovery synergistic control strategies for urban rail. Adithya et al. [21] revealed significant meteorological impacts on Swedish railway delays through extreme weather event analyses, while William et al. [22] demonstrated strong correlations between abrupt weather changes and delay propagation. Zhou et al. [23] tackled scheduling complexities in high-speed rail emergency scenarios (e.g., strong winds, foreign object collisions) by proposing a parallel railway traffic management (RTM) system based on the ACP framework (Artificial Systems-Computational Experiments-Parallel Execution). Through agent-based modeling of artificial RTM environments and multi-objective optimization strategies (hybrid, FCFS, FSFS), the system demonstrated superior train rescheduling capabilities via real-time physical-artificial system feedback loops in temporary speed restriction and complete blockage scenarios, outperforming traditional strategies in emergency response efficiency and dispatcher decision support. Zhou et al. [24] integrated GIS high-resolution precipitation data with non-spatial high-speed rail operation data to construct a grid model. Empirical analysis of the 2015–2017 rainy seasons in eastern China revealed that extreme rainfall significantly exacerbated daily surface rainfall delays on Hangzhou-Shenzhen and Nanjing-Hangzhou lines, with Beijing-Shanghai lines

more sensitive to rainfall intensity and Shanghai-Nanjing/Denver-Wenzhou lines most vulnerable to extreme precipitation. This led to regional adaptive strategies for enhancing climate resilience in high-speed rail systems. Wang et al. [25] proposed a dual-layer model predictive control (MPC) framework for high-speed rail online delay management and train control. The upper layer optimized global train delay minimization, while the lower layer coordinated operational time constraints with energy efficiency objectives. Validation using Beijing-Shanghai High-Speed Railway data demonstrated significant improvements in real-time performance, delay reduction efficiency, and robustness against multi-disturbance scenarios compared to FCFS/FSFS benchmarks. Song et al. [26] developed an autonomous route management system based on colored Petri nets, verifying its safety and performance to enhance station delay handling efficiency. Song et al. [27] proposed an autonomous train control system that improves train operation coordination and delay handling capabilities through data fusion and predictive control.

Despite these advancements, conventional methods remain constrained by static rules or single-scenario assumptions, exhibiting notable shortcomings in dynamic modeling, multi-objective real-time trade-offs, and high-dimensional constraint solving efficiency. In recent years, machine learning techniques have demonstrated transformative potential in addressing these challenges: Chen et al. [28] developed a deep learning model that effectively captures complex spatiotemporal correlations, significantly improving delay prediction accuracy. Luo et al. [29] introduced a Bayesian-optimized multi-output model for dynamic parameter adjustment, enhancing sequential train delay assessment and real-time forecasting capabilities. Shady et al. [30] empirically validated the adaptability and practicality of machine learning in complex railway scenarios through real-world deployment. Sun et al. [31] addressed the challenges of electromagnetic suspension systems in maglev trains under complex operational conditions such as track irregularities, external disturbances, time-varying mass, and input delays. They proposed an adaptive neural network controller integrating input delay compensation and parameter optimization. This approach employs a dual-layer neural network to approximate uncertain dynamics, a sliding mode surface delay compensation design, and Actor-Critic reinforcement learning for real-time parameter optimization. Lyapunov theory was used to prove finite-time stability, with simulations and experiments demonstrating superior performance in suppressing air-gap vibrations caused by delays and uncertain dynamics, significantly outperforming traditional methods and enhancing suspension control efficiency. Yue et al. [32] tackled the real-time train timetable reorganization (TTR) challenge in high-speed rail by introducing a multi-stage decision-making framework based on reinforcement learning. The framework optimizes training efficiency through a compact, high-quality action set and uncertainty-aware action sampling strategies while designing a rule-free scheduling policy self-learning mechanism. Experimental validation confirmed its universality and competitiveness across diverse scenarios, establishing a novel paradigm for intelligent scheduling under dynamic disruptions. Wang et al. [33] addressed the issue of traction power consumption accounting for 50% of total metro operational energy, proposing an energy-saving deep reinforcement learning algorithm (ES-MEDRL) that integrates Lagrange multipliers and maximum policy entropy. By constructing a dual-objective optimization function with enhanced velocity domain exploration and a quadratic time-energy trade-off strategy, the algorithm achieved a 20% reduction in traction energy consumption compared to manual driving on the Yizhuang Metro Line in Beijing. It simultaneously balanced operational comfort, punctuality, and safety, offering a new paradigm for intelligent energy-efficient scheduling at the metro system planning level. Qiao et al. [34] addressed challenges in millimeter-wave communication for high-speed rail, including rapid time-varying channel modeling and beam management. Their intelligent beam management scheme based on deep Q-networks

(DQN) exploits hidden patterns in millimeter-wave train-to-ground communication systems, improving downlink signal-to-noise ratio (SNR) while ensuring communication stability and low training overhead. Simulations confirmed its superior performance over four baseline methods, highlighting advantages in SNR stability and implementation complexity. Ling et al. [35] focused on lightweight, high-quality data transmission and dynamic interaction requirements for sensor monitoring and remote communication in future intelligent high-speed rail networks. They proposed a self-powered multi-sensor monitoring and communication integration system, featuring a low-power backscatter communication framework, Gaussian mixture model analysis for coverage regions, and a total task completion time optimization problem considering energy transfer, data collection, and transmission rate constraints. An innovative option-based hierarchical deep reinforcement learning method (OHDRL) was developed to address system complexity, with experiments showing significant improvements in reward values and learning stability over existing algorithms. These advancements establish a theoretical foundation for the integration of intelligent algorithms and dynamic modeling. However, developing scheduling optimization methods that simultaneously achieve forward-looking design, robustness, and real-time capability remains a critical challenge for addressing multidimensional uncertainties in high-speed rail systems under adverse weather conditions.

In summary, existing research has made significant progress in delay propagation modeling, multi-objective optimization, and collaborative scheduling. However, three shortcomings persist in snowstorm scenarios: First, most methods rely on static rules or single-scenario assumptions, failing to capture the dynamic propagation characteristics of extreme weather. Second, achieving real-time trade-offs among safety constraints, on-time performance, and scheduling stability remains challenging. Third, solution efficiency is limited under high-dimensional constraints. Although deep learning and reinforcement learning demonstrate potential in delay prediction and dynamic scheduling, constructing a scheduling optimization framework that integrates foresight, robustness, and real-time capability remains a core challenge. To address these issues, this paper proposes an LSTM-PPO-based dynamic scheduling optimization algorithm for high-speed rail. This method leverages LSTM networks to capture long-term dependencies in snowstorm propagation and delay diffusion, while employing the PPO algorithm to ensure stable policy updates. By simulating snowstorm conditions, it predicts speed-restricted sections and overhead contact system failure risks, ultimately establishing a dynamic scheduling strategy for high-speed rail that combines real-time responsiveness with adaptability to complex scenarios. This approach effectively addresses the limitations of traditional methods in snowstorm response, providing new theoretical support for high-speed rail dynamic scheduling.

## 2. Problem Description and Modeling

### 2.1. Problem Description

In real-world operational scenarios, train timetables are typically formulated by relevant railway authorities. When encountering heavy snowstorms, high-speed rail (HSR) systems are subjected to multiple dynamic disturbances, triggering cascading effects such as speed restrictions, delays, and even operational strategy adjustments (as shown in Figure 1). The inherent uncertainties in HSR operational environments pose significant challenges when actual conditions deviate markedly from preset parameters or when exceptional scenarios exceed predefined rules. Conventional methods often struggle to deliver timely and flexible responses under such circumstances. Consequently, the LSTM-PPO algorithm proposed in this study demonstrates superior performance over traditional approaches in dynamic scheduling under snowstorm-induced disruptions, establishing a robust foundation for real-time HSR scheduling during emergencies.

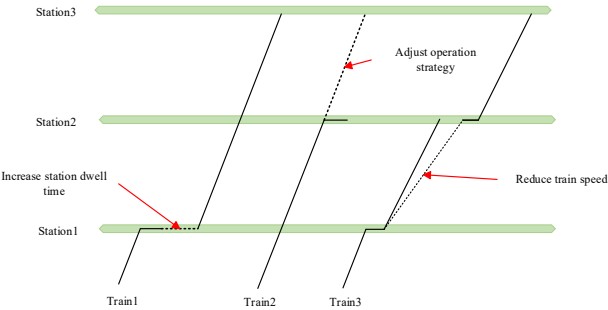

**Figure 1.** Train operation interference diagram.

### 2.2. Model Development

To minimize the total delay duration at each station and reduce overall operational disruption caused by blizzard-induced train delays while satisfying passenger timeliness requirements, the proposed multi-objective optimization model is formulated as follows:

$$MinF_1 = \sum_{i \in G} \sum_{j=1}^{\Omega} \left| A_{i,j}^0 - T_{i,j}^0 \right| + \sum_{i \in G} \sum_{j=1}^{\Omega} \left| A_{i,j}^1 - T_{i,j}^1 \right| \tag{1}$$

$$MinF_2 = \sum_{i \in G} \sum_{j=1}^{\Omega} \left| A_{i,j}^1 - A_{i,j}^0 \right| - \sum_{i \in G} \sum_{j=1}^{\Omega} \left| T_{i,j}^1 - T_{i,j}^0 \right| \tag{2}$$

$MinF_1$ measures the deviation of actual arrival and departure times at each station from the scheduled timetable, reflecting train punctuality; $MinF_2$ quantifies the difference between actual stopping or passing times and the planned schedule, evaluating the disruption level caused by dispatching adjustments, thereby avoiding frequent modifications that may lead to operational chaos and reflecting the stability of train scheduling.

The parameter definitions and decision variable meanings in this paper are presented in Tables 1 and 2.

**Table 1.** Parameters used and their definitions.

| Symbol Parameters | Parameter Meaning | Symbol Parameters | Parameter Meaning |
|---|---|---|---|
| $G$ | Set of Trains Requiring Dynamic Scheduling | $T_{i,j}^1$ | Scheduled Departure Time of Train $i$ from Station $j$ |
| $M$ | Set of All Trains in the Scheduled Operation Network | $C_j$ | Total Number of Tracks at Station $j$ |
| $\Omega$ | Number of Stations Passed by the Operating Route | $N_{i,j}$ | Track Occupancy of Train $I$ at Station $j$ |
| $i$ | Train ID, $i = 1, 2, \cdots, G$ | $O_j$ | Represents the Track Occupancy Ratio at Station $j$ |
| $j$ | Station ID, $j = 1, 2, \cdots, \Omega$ | $\Delta S_{i,(j,j+1)}^{(e_k)}$ | Additional Travel Time of Train $I$ on Section $(j, j+1)$ under the Impact of Disruption $e_k$ |
| $L_{(j,j+1)}$ | Length of Section $(j, j+1)$ | $Z_{i,(j,j+1)}$ | Actual Travel Speed of Train $i$ on Section $(j, j+1)$ after Dynamic Adjustment |
| $T_{i,j}^0$ | Scheduled Arrival Time of Train $i$ at the Station | $Z_{max}$ | Maximum Actual Speed of the Train |
| $\Delta T_{i,(j,j+1)}^{(e_k)}$ | Additional maintenance time due to contact with power lines, icing, or equipment failure | $\Delta \hat{S}_{i,j}$ | Predicted Delay Increment |
| $T_{repair}$ | Maintenance Period | $v_{max}$ | Maximum Permitted Operating Speed for Section |

**Table 2.** Decision variables and their definitions.

| Decision Variables | Meaning |
| --- | --- |
| $A_{i,j}^0$ | Actual Arrival Time of Train $i$ at Station $j$ after Dynamic Adjustment |
| $A_{i,j}^1$ | Actual Departure Time of Train $i$ from Station $j$ after Dynamic Adjustment |

The constraint conditions for the model are formulated as follows:

(1) Section operation and safety spacing, the arrival time of train $i$ at station $j+1$, after departing from station $j$, should account for the original travel time $S_{i,(j,j+1)}$, potential event-induced delays $\Delta S_{i,j}^{(e_k)}$, as well as the dwell or passing time $D_{i,j}$ at station $j$. That is:

$$A_{i,(j+1,j+2)}^0 + A_{i,(j+1,j+2)}^1 \geq A_{i,(j,j+1)}^0 + A_{i,(j,j+1)}^1 + D_{i,(j,j+1)} + \frac{L_{(j,j+1)}}{Z_{i,(j,j+1)}} + \Delta S_{i,(j,j+1)}^{(e_k)}, \forall i \in M, j = 1, \ldots, \Omega \tag{3}$$

(2) Minimum Running Time Between Sections, ensuring that trains comply with minimum dwell (or running) time requirements at stations (or specific sections), and meet safety or technical specifications, i.e.,

$$D_{i,j} \geq L_{i,j}, \forall i \in M, j = 1, \ldots, \Omega \tag{4}$$

(3) Trains must not depart or arrive earlier than the scheduled time. If there are mandatory regulations prohibiting early arrival (or early departure), this constraint is required, otherwise passengers may miss their trains or cause station connection conflicts, i.e.,

$$A_{i,j}^0 \geq T_{i,j}^0, A_{i,j}^1 \geq T_{i,j}^1, \forall i \in M, j = 1, \ldots, \Omega \tag{5}$$

(4) Avoiding station capacity and occupancy conflicts: When two trains are at the same platform or section, it must be ensured that occupancy conflicts do not occur. This constraint can be formalized as the station accommodating a limited number of trains simultaneously within the same time period, i.e.,

$$\frac{L_{(j,j+1)}}{Z_{i,(j,j+1)}} \geq \frac{L_{(j,j+1)}}{Z_{\max}}, A_{i,j}^0 \neq A_{k,j}^0, \forall i \in M, \forall i \neq k, j = 1, \ldots, \Omega \tag{6}$$

(5) Track Occupancy Constraint: When the track occupancy ratio at a station exceeds 90%, incoming train operations are prohibited until the occupancy rate drops below the threshold, i.e.,

$$O_j = \frac{N_{i,j}}{C_j} \leq 0.9 \tag{7}$$

## 3. Strategy Optimization Based on the LSTM-PPO Algorithm

*3.1. MDP (Markov Decision Process) Modeling*

MDP serves as the problem modeling framework for reinforcement learning, providing a formal abstraction for high-speed rail scheduling by defining the following core components: state, action, and reward. This approach transforms the complex high-speed rail scheduling problem into a structured 'state-action-reward' framework, ensuring the mathematical rigor of the decision-making logic.

3.1.1. State Space Design

In high-speed rail dynamic dispatching optimization, the design of the state space is the foundational basis for reinforcement learning models to perceive environmental

dynamics and implement intelligent decision-making. To address the core requirements of speed restrictions, station stop adjustments, and dynamic track occupancy dispatching, this study redefines the state space by focusing on three key components: real-time operational status, environmental perturbation parameters, and historical dependency features.

Real-time operational status includes actual arrival and departure times ($A_{i,j}^0$, $A_{i,j}^1$), reflecting the degree of train operation deviation; station dwell time adjustment ($\Delta D_{i,j}$), recording dynamically adjusted dwell times; track occupancy rate ($O_j$) and track occupancy status ($G_{i,j}$), describing station resource competition. Environmental perturbation parameters include section snowstorm level ($v_{(j,j+1)}$), overhead contact line icing status ($I_{(j,j+1)}$), and track friction coefficient ($\mu_{(j,j+1)}$), quantifying the impact of weather on operational safety. Historical dependency features include the LSTM hidden state ($h_t$), encoding historical temporal features to capture long-term dependencies in delay propagation; section disturbance time ($\Delta S_{i,j}^{(e_k)}$), characterizing additional travel delays caused by unexpected events; and predicted delay times ($\Delta S_{i,j}^{(t+1:t+T)}$) for each section within the next T time steps, forecasted via LSTM and incorporated as part of the PPO state space to provide the agent with forward-looking information. In summary, the state space is defined as the set $s_t = \left\{ A_{i,j}^0, A_{i,j}^1, \Delta S_{i,j}^{(e_k)}, v_J, I_J, O_{i,j}, G_{i,j}, h_t, \Delta S_{i,j}^{(e_k)}, \Delta S_{i,j}^{(t+1:t+T)} \right\}$.

3.1.2. Action Space Design

To effectively address the multiple uncertainties caused by snowstorm-induced train delay propagation and dynamic safety constraints, the action space must encompass key scheduling operations while satisfying multidimensional constraints related to real-time responsiveness, safety, and feasibility. Traditional methods often rely on static rules or single-action modes, such as fixed speed restriction ratios, which struggle to dynamically balance the trade-off between punctuality rates and safety risks. Additionally, in complex railway network scenarios, these approaches are prone to causing action dimension explosion or policy oscillation. The action space defined in this study comprises discrete and hybrid decisions, aiming to achieve multi-objective collaborative optimization through a finite and structured set of operations. The specific scheduling instructions executable by the agent include speed restriction grade adjustment ($\alpha_j$), to align with changes in snowstorm severity; dynamic adjustment of station dwell times ($\Delta D_{i,j}$), to regulate resource competition; and action ($\alpha_t$), which describes the agent's choice between speed restriction or station stop strategies.

Under snowstorm conditions, the speed restriction strategy for high-speed trains must be dynamically aligned with snowstorm severity to balance safety and operational efficiency. The specific mapping rule is defined as shown in Equation (8): the higher the snowstorm severity level $v_{(j,j+1)}$, the lower the maximum allowable speed ratio $a_{(j,j+1)}$. Here, $v_{(j,j+1)}$ denotes the snowstorm severity level of section $(j, j + 1)$.

Dynamic adjustment of station dwell time is one of the key decision variables in high-speed rail dynamic dispatching. It involves elastically modifying the actual dwell time at stations based on real-time operational status and resource constraints. The dynamic adjustment is formulated as follows:

$$\Delta D_{i,j} = \eta \cdot \left( \frac{\Delta T_{i,(j,j+1)}^{(e_k)}}{T_{repair}} \right) + \beta \cdot \left( \frac{v_{(j,j+1)}}{v_{\max}} \cdot \left| A_{i,j}^0 - T_{i,j}^0 \right| \right) \tag{8}$$

When overhead contact system icing occurs ($I_{(j,j+1)} = 1$), $D_{i,(j,j+1)} \geq L_{i,(j,j+1)} + T_{repair}$ must be strictly adhered to prioritize safety.

The $\Delta S_{i,j}^{(t+1:t+T)}$ (predicted delay) and $\sigma_{i,j}^{(t)}$ (confidence level) output by the LSTM prediction module are used as input conditions. The pre-action triggering rule can be defined as follows: if the predicted delay $\Delta S_{i,j}^{(t+1:t+T)}$ for a specific section $(j, j+1)$ satisfies $\Delta \hat{S}_{i,j} \geq \Delta S_{threshold}$ (where $\Delta S_{threshold}$ is set to the 95th percentile of historical delays), the pre-action mechanism is triggered. The speed restriction ratio for the section is then pre-adjusted in advance steps to satisfy: $\alpha_j = \max\left(0.3, \alpha_j^{current} - \gamma \cdot \frac{\Delta \hat{S}_{i,j}}{\Delta S_{max}}\right)$ where $\gamma$ is the attenuation coefficient and $\Delta S_{max}$ is the maximum allowable delay increment.

A pre-action flag $b_j \in (0, 1)$ is defined to indicate whether the pre-action is triggered. The action generation rule is formulated as Equation (9). If the pre-action conflicts with the current state, the pre-action is forcibly canceled, and dynamic station dwell time correction is triggered.

$$\alpha_t = \begin{cases} Speed\ restriction\ adjustment & b_j = 1\ and \Delta \hat{S}_{i,j} \geq \Delta S_{threshold} \\ Original\ action\ rule & else \end{cases} \quad (9)$$

3.1.3. Reward Function Design

In reinforcement learning algorithms, the design of the reward function serves as a critical mechanism to drive the agent toward optimal decision-making under multi-objective trade-offs. Traditional approaches often rely on single-objective optimization or static weight allocation, which struggle to dynamically balance complex trade-offs between punctuality rates and scheduling stability during snowstorm conditions, often leading to suboptimal strategies. To address this, this study proposes a weighted multi-objective reward function that dynamically integrates punctuality rewards and scheduling stability to guide the agent toward collaborative optimization in partially observable environments. The following sections will elaborate on the mathematical modeling of individual reward components, the logic for weight allocation, and the collaborative optimization mechanism.

$$R_t = \omega_1 R_{time} + \omega_2 R_{stability} + \omega_3 R_{occupancy} + \omega_4 R_{predict} + \omega_5 R_{proactive} \quad (10)$$

The following subsections present the detailed design and descriptions of each component:

①. Punctuality Reward: It is calculated as the base punctuality rate minus the relative deviation ratio, incorporating a Sigmoid function for smoothing.

$$R_{time} = \sum_{i,j} \left[ \frac{1}{1 + \left| \frac{A_{i,j}^0 - T_{i,j}^0}{T_{max}} \right|} + \frac{1}{1 + \left| \frac{A_{i,j}^1 - T_{i,j}^1}{T_{max}} \right|} \right] \quad (11)$$

②. Scheduling Stability Penalty: Based on the maximum allowable adjustment amplitude, the penalty is calculated as an inversely proportional function of the adjustment amplitude, incorporating a quadratic term to amplify the penalty for large adjustments. This suppresses over-optimization of short-term punctuality, ensuring that larger adjustment amplitudes result in stronger penalties.

$$R_{stability} = \sum_{i,j} \left[ 1 - \frac{\left| \left(A_{i,j}^1 - A_{i,j}^0\right) - \left(T_{i,j}^1 - T_{i,j}^0\right) \right|}{\Delta D_{max}} \right]^2 \quad (12)$$

③. Track Occupancy Rate Penalty: To prevent resource contention caused by excessive track occupancy, a linear penalty is applied when the track occupancy rate exceeds 80%.

For each 1% exceeding this threshold, a penalty of 1 unit is imposed, thereby suppressing resource contention risks.

$$R_{occupancy} = -\sum_j \Pi(O_j \geq 0.8) \cdot (O_j - 0.8) \tag{13}$$

④. Prediction Accuracy Reward: Constrains the LSTM prediction error to enhance prediction reliability. The error penalty is calculated only for sections where delays actually occur, avoiding interference from invalid predictions in the strategy.

$$R_{predict} = -\sum_j \left| \Delta \hat{S}_{i,j}^{(t)} - \Delta S_{i,j}^{(t)} \right| \cdot \Pi\left( \Delta \hat{S}_{i,j}^{(t)} > 0 \right) \tag{14}$$

⑤. Proactive Reward Item: The proactive reward item aims to encourage actions that prevent the spread of delays in advance. If action $\alpha_t$ results in the subsequent actual delay satisfying $\Delta S_{i,j}^{(t+k)} \leq x \Delta \hat{S}_{i,j}^{(t+k)}$, it is deemed effective in alleviating delays. The value of the $x \in [0,1]$ agent is determined based on its learning process. The reward calculation can be defined as follows: a reward of +1 is granted for each successful prevention of a significant delay.

$$R_{proactive} = \sum \Pi\left( \Delta \hat{S}_{i,j}^{(t+k)} \geq \Delta S_{threshold} \text{ and } \Delta S_{i,j}^{(t+k)} \leq x \Delta \hat{S}_{i,j}^{(t+k)} \right) \tag{15}$$

⑥. Dynamic Weight Adjustment Mechanism: To address the changing priorities of objectives in different scenarios, an adaptive weight allocation mechanism is designed. If a certain objective performs poorly, its weight is automatically increased to prioritize the optimization of that objective.

$$\omega_i' = \frac{\omega_i \cdot \exp(-\eta R_i)}{\sum \omega_i \cdot \exp(-\eta R_i)} \tag{16}$$

Through the above design, the reward function can dynamically balance multi-objective conflicts, providing theoretical guarantees for the stable training of the LSTM-PPO algorithm in complex scenarios.

*3.2. PPO Algorithm Design*

PPO (Proximal Policy Optimization) is a reinforcement learning algorithm based on the Actor-Critic framework [34]. Its core lies in achieving collaborative optimization through separated Actor and Critic networks. The Actor network is responsible for generating action policies (denoted as $\pi_\theta(\alpha|s)$), directly controlling scheduling commands such as speed limit adjustments and dwell time corrections. The Critic network evaluates the state value (denoted as $V_\theta(s)$), predicts the long-term cumulative reward, and guides the Actor in optimizing the direction of action selection. In the context of high-speed rail dynamic scheduling, the advantages of the Actor-Critic framework are evident: the Actor focuses on action generation, while the Critic focus on state evaluation, avoiding conflicts arising from a single network handling multiple objectives. Additionally, the Clip mechanism constrains the magnitude of the Actor's policy updates, and combined with the variance penalty of the Critic's value function, it suppresses policy oscillation.

The Actor network is updated through policy gradient optimization to refine the action probability distribution. The speed limit ratio $a_j$ and dwell time correction $\Delta D_{i,j}$ are incorporated as conditional inputs to the policy distribution $\pi_\theta$, enabling the Actor network to generate actions that directly respond to speed restriction demands and dwell time adjustment constraints under heavy snow weather. When the LSTM predicts a significant future delay j $\Delta \hat{S}_{i,j}$ for a specific section with high confidence, the weight $\sigma(\Delta \hat{S}_{i,j})$ is increased, prompting the policy to prioritize adjustments to the dwell time $\Delta D_{i,j}$ or speed

limit ratio $a_j$ for that section. This weight allocation coordinates the conflicting objectives of punctuality, stability, and safety:

$$L^{CLIP}(\theta) = E_t\left[\min\left(\frac{\pi_\theta(\alpha|s)}{\pi_{old}(\alpha|s)}\hat{A}_t, clip(r_t(\theta), 1-\varepsilon, 1+\varepsilon)\hat{A}_t\right)\right] \tag{17}$$

$$r_t(\theta) = \frac{\pi_\theta(\alpha_t|s_t, a_j, \Delta D_{i,j})}{\pi_{old}(\alpha_t|s_t)} \tag{18}$$

$$\hat{A}_t = \sum_j \sigma(\Delta\hat{S}_{i,j}) \cdot \left(\omega_1 \cdot \delta\left(A_{i,j}^0 - T_{i,j}^0\right) + \omega_2 \cdot \delta\left(\left(A_{i,j}^1 - A_{i,j}^0\right) - \left(T_{i,j}^1 - T_{i,j}^0\right)\right) + \omega_3 \cdot \log(O_j)\right) \tag{19}$$

The Critic network is updated by optimizing the composite value function loss to improve the estimation of state value. The second term suppresses the variance in the value function, while the third term introduces a penalty for LSTM prediction errors, achieving collaborative optimization of scheduling stability and prediction accuracy. The target return is calculated using a discount factor $\gamma$, balancing immediate rewards and long-term gains. The objective function is formulated as follows:

$$L^{VF}(\theta) = E_t\left[\left(V_\theta(s_t) - R_t^{target}\right)^2\right] + \lambda_1 \cdot Var(V_\theta(s_t)) + \lambda_2 \cdot \sum\left|\Delta\hat{S}_{i,j} - \Delta S_{i,j}\right| \tag{20}$$

$$R_t^{target} = \sum_{k=0}^{T-t} \gamma^k r_{t+k} + \gamma^{T-t}V_\theta(s_T) \tag{21}$$

The Actor adjusts its policy direction based on the advantage signal $\hat{A}_t$ provided by the Critic, while the Critic optimizes its value estimation relying on the action distribution of the Actor. The two components are updated asynchronously to reduce coupling oscillations.

*3.3. LSTM-PPO Algorithm Architecture Design*

3.3.1. LSTM Algorithm Architecture Design

In the LSTM-PPO algorithm, the core role of the LSTM network is to act as a state encoder, specifically designed to handle the inherent temporal dependencies in train scheduling problems. It is capable of extracting and compressing historical information from a sequence of consecutive state observations, providing a higher-dimensional feature representation that incorporates contextual information for the subsequent decision-making networks (the Actor-Critic component of PPO) [35].

The input to the LSTM consists of two parts: the external input at the current time step and the internal state from the previous time step. The state vector, observed at each decision time step $t$, is a multi-dimensional feature vector that includes static attributes such as train identifiers, station topology, and preset train schedules, as well as dynamic variables such as the train's current actual delay, real-time speed, distance to the preceding train, and external environmental factors. Additionally, expert information is included during the imitation learning phase, with expert decisions forming part of the input. The previous hidden state $c_{t-1}$ and cell state $h_{t-1}$ represent the internal memory units of the LSTM network. $h_{t-1}$ is the output from the previous time step, while $c_{t-1}$ serves as the "long-term memory" that persists throughout the entire sequence. By combining these two vectors with the current state $s_t$, the LSTM is able to integrate historical information with the present state, thereby capturing the dynamic evolution of the system state.

Correspondingly, the output of the LSTM also consists of two parts. The context feature vector $o_t$ is the primary output of the LSTM at the current time step t. It is a high-dimensional feature embedding that integrates the current input $s_t$ with its historical context $h_{t-1}$ and $c_{t-1}$ through a nonlinear combination. This vector is not the final decision but

serves as input for subsequent networks. Its advantage lies in the fact that the Actor network utilizes $o_t$ to compute the policy distribution $\pi_\theta(\alpha|s)$, which represents the probabilities of taking various actions under the current state; meanwhile, the Critic network employs $o_t$ to estimate the value $V_\theta(s)$ of the current state, representing the expected return achievable by following the current policy from the current state onward. The hidden state $h_t$ and cell state $c_t$ at the current time step function as internal memory units, which are passed to the next time step t + 1, thereby enabling the continuous flow of information across the time sequence.

In the reinforcement learning paradigm, the dataset required by the LSTM is dynamically generated through the interaction between the agent and the environment. The National Railway Administration collects foundational static data, such as train initial schedules, to initialize the simulation environment. However, the actual training data consists of experience tuples, represented in the form $(s_t, a_t, r_t, s_{t+1}, d_t)$, where $s_t$ denotes the current state, $a_t$ is the action taken by the agent, $r_t$ is the reward provided by the environment, $s_{t+1}$ is the next state feedback from the environment, and dd is a flag indicating whether the state is terminal. These experience tuples are collected by running thousands of episodes of the agent within the simulation environment, forming a large experience pool or rolling trajectory.

This study adopts an advanced strategy of curriculum learning to organize the generation of training data. In the early stages of training, the environment is relatively simple; as training progresses, the difficulty of the environment is gradually increased. This data generation approach, which transitions from simple to complex, enables the model to first learn basic scheduling strategies and then progressively generalize to more complex and uncertain environments, significantly improving the stability of training and the robustness of the final model.

### 3.3.2. Dual-Module Collaborative Mechanism

To address the complex spatiotemporal dynamic characteristics of high-speed rail scheduling under heavy snow weather, this paper proposes a separated dual-module framework, consisting of an LSTM prediction module and a PPO decision module, to achieve collaborative optimization between prediction and decision-making. In the LSTM module, historical operation status sequences $s_{t-T:t}$, real-time heavy snow levels $v_j$, and future weather prediction data $W_{t+1:t+T}$ are input, and the outputs include the additional travel time $\Delta\hat{S}_{i,j}^{(t+1:t+T)}$ for each section within the next TT time steps and its confidence interval $\left[\Delta\hat{S}_{low}, \Delta\hat{S}_{high}\right]$. The loss function adopts Huber loss to balance prediction accuracy and robustness, avoiding interference from outliers, where $\delta = 1.0$ is the smoothing threshold and $\Delta S$ represents the actual delay time.

$$L_{pred} = \sum \delta \left( \left| \Delta\hat{S} - \Delta S \right| - \frac{\delta}{2} \right) \quad (22)$$

In the PPO module, the current state $s_t$, which includes real-time operational data, environmental parameters, and the LSTM prediction results $\Delta S_{i,j}^{(t+1:t+T)}$, is input. The predictive information and real-time state are encoded into a joint feature vector through a shared hidden layer, where $W_p$ and $W_d$ represent the weight matrices for the prediction module and state encoding, respectively. The ReLU activation function enhances the nonlinear expressive capability.

$$z_t = ReLU\left(W_p h_t^{pred} + W_d h_t^{sate}\right) \quad (23)$$

To address the impact of predictive uncertainty on the policy, a confidence-driven dynamic weight allocation strategy is designed. Based on the predictive error variance $\sigma_{i,j}^{(t)} = Var\left(\Delta \hat{S}_{i,j}^{(t+1:t+T)}\right)$, the weight $w_{pred} = \frac{1}{1+\sigma_{i,j}^{(t)}}$ of the predictive information in the policy network is dynamically adjusted. When the prediction confidence is high, the policy network prioritizes reliance on the predictive results; conversely, when confidence is low, the predictive weight is reduced to avoid misleading decisions.

The synchronous update of the LSTM prediction module and the PPO decision module may lead to gradient conflicts, which can decrease model convergence efficiency. To avoid gradient conflicts between the prediction module and the policy module, a phased asynchronous update mechanism is adopted. The PPO decision module is updated every 50 steps through the policy loss $L^{CLIP}(\theta)$ and the value function loss $L^{VF}(\theta)$, ensuring policy stability. The LSTM prediction module is updated every 10 steps based on $L_{pred}$, focusing on modeling temporal dependencies. The specific architecture is illustrated in Figure 2.

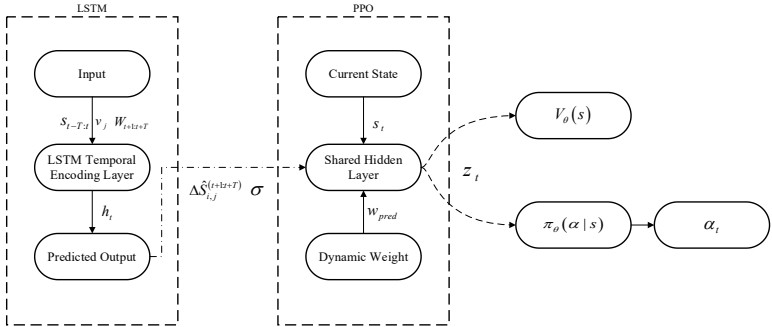

**Figure 2.** LSTM-PPO algorithm structure diagram.

## 4. Simulation Experiments and Results Analysis

### 4.1. Experimental Environment Design

This study selects the Lanzhou-Xinjiang High-Speed Railway (Lanzhou-Xinjiang Passenger Dedicated Line) section from Lanzhou West Station to Urumqi Station as the experimental validation object. This line is characterized by a prominent high-cold climate, with an average winter operating temperature below $-15\ °C$, a maximum wind speed of 30 m/s, and an annual average of over 20 days of heavy snowfall. The operational stability of the line faces multiple meteorological threats, including dynamic changes in snow depth, high probability of contact wire icing, and abrupt reductions in visibility.

According to the disaster classification criteria in this study, when the snowfall intensity on the core sections of the line exceeds the threshold of 0.6, the track friction coefficient decreases by 60%, triggering a speed restriction mechanism of 200 km/h. When the blizzard intensity surpasses the threshold of 0.8, the probability of contact wire icing surges to 50%, accompanied by visibility dropping below 50 m, necessitating an emergency speed restriction of 150 km/h. These meteorological conditions lead to compound risks, such as degradation of train dynamic performance, extended braking distances, and instability of the power supply system, posing severe challenges to the real-time risk perception and multi-objective coordination capabilities of the dynamic scheduling system. This provides a high-value experimental scenario for validating the adaptability of the LSTM-PPO algorithm in key aspects such as blizzard propagation modeling, contact wire icing warning, and emergency speed restriction decision-making.

The specific engineering parameters and operational characteristics of this section are detailed in Table 3. These parameters include the line length, number of interval

stations, average distance between stations, and major geographical environmental features, providing comprehensive benchmark data for model performance evaluation.

**Table 3.** Detailed parameters of the Lanzhou-Xinjiang High-Speed Railway from Lanzhou West Station to Urumqi Station.

| Section | Running Time/min | Dwell Time/min | Section Distance/km |
|---|---|---|---|
| Lanzhou West Station—Haidong Station | 10:25–11:07 | 2 | 132 |
| Haidong Station—Xining Station | 11:09–11:29 | 6 | 56 |
| Xining Station—Minle Station | 11:35–12:58 | 2 | 233 |
| Minle Station—Zhangye West Station | 13:00–13:25 | 3 | 64 |
| Zhangye West Station—Lizhai South Station | 13:28–13:44 | 2 | 35 |
| Lizhai South Station—GaoTai South Station | 13:46–14:03 | 2 | 36 |
| GaoTai South Station—Jiuquan South Station | 14:05–14:47 | 2 | 126 |
| Jiuquan South Station—Jiayuguan South Station | 14:49–15:00 | 6 | 21 |
| Jiayuguan South Station—Yumen Station | 15:06–15:51 | 14 | 126 |
| Yumen Station—Liuyuan South Station | 16:05–17:11 | 2 | 163 |
| Liuyuan South Station—Hamih Station | 17:13–18:44 | 6 | 261 |
| Hamih Station—Shanshan North Station | 18:50–20:15 | 2 | 281 |
| Shanshan North Station—Turpan North Station | 20:17–20:45 | 2 | 91 |
| Turpan North Station—Urumqi Station | 20:47–21:57 | 27 | 167 |

### 4.2. Model Parameter Settings

Prior to the formal training of the LSTM-PPO model, a series of parameter sensitivity analyses and hyperparameter optimization experiments were conducted to determine the optimal model configuration. These parameter tuning experiments were based on a systematic evaluation of model performance, employing a progressive and adaptive parameter adjustment strategy. The model utilizes a dynamic parameter adjustment mechanism, where the clipping range parameter follows a decremental strategy, progressively narrowing as training progresses to enhance policy stability and promote convergence. By comparing cumulative rewards, policy loss, and value function loss under different parameter configurations, the optimal parameter settings presented in Table 4 were obtained.

**Table 4.** Parameters configuration of the LSTM-PPO model.

| Value Range | Value |
|---|---|
| GAMMA | 0.999 |
| LAMBDA | 0.98 |
| LR | $1 \times 10^{-5}$ |
| BATCH_SIZE | 64 |
| NUM_WORKERS | 8 |
| MAX_EPOCHS | 1000 |
| ROLLOUT_STEPS | 64 |
| PPO_EPOCHS | 15 |
| CLIP_EPSILON | 0.01 |
| MAX_GRAD_NORM | 0.1 |
| POLICY_WEIGHT | 1.2 |
| VALUE_WEIGHT | 0.05 |
| ENTROPY_WEIGHT | 0.0005 |
| INPUT_SIZE | 8 |
| HIDDEN_SIZE | 256 |
| NUM_LAYERS | 2 |

### 4.3. Sensitivity Analysis

To validate the robustness of the multi-objective reward function in the LSTM-PPO model and guide weight parameter optimization, this study conducted a systematic sensitivity analysis on five key weight parameters. A grid search method tested 80 different weight combinations, evaluating the impact and interactions of each weight parameter on model performance. The results are shown in Table 5.

**Table 5.** Weight Sensitivity Analysis Results.

| Weight Type | Baseline Value | Test Range | Optimal Value | Sensitivity Score | Performance Impact | Correlation Coefficient |
|---|---|---|---|---|---|---|
| Punctuality Reward | 0.25 | 0.15–0.35 | 0.596 | 19.91 | +15.2% | 0.831 |
| Scheduling Stability Penalty | 0.15 | 0.10–0.25 | 0.138 | 14.91 | +8.5% | 0.645 |
| Track Occupancy Rate Penalty | 0.20 | 0.15–0.30 | 0.158 | 8.73 | +6.3% | 0.512 |
| Prediction Accuracy Reward | 0.25 | 0.20–0.35 | 0.058 | 5.55 | −4.2% | −0.328 |
| Proactive Reward Item | 0.15 | 0.05–0.25 | 0.050 | 4.06 | −3.1% | −0.287 |

Experimental results indicate that the prediction accuracy reward weight is the most critical factor influencing system performance, with a sensitivity score as high as 19.91. It exhibits a strong positive correlation (r = 0.831) with overall performance, meaning even minor weight adjustments can significantly impact the model's overall performance. The weights for the forward-looking reward and on-time reward exhibit moderate sensitivity, both positively correlated with performance, indicating these reward mechanisms play a vital role in enhancing system performance. In contrast, the track occupancy penalty and scheduling stability penalty weights demonstrate low sensitivity and negative correlations with performance, suggesting excessive penalty mechanisms may suppress the system's learning effectiveness.

The optimal weight configuration identified through sensitivity analysis significantly outperforms the baseline: the prediction accuracy reward weight increases substantially from 0.25 to 0.596, the forward reward weight is moderately reduced to 0.138, the punctuality reward weight is fine-tuned to 0.158, while both penalty weights are substantially decreased. This configuration embodies an optimization strategy of "reward-driven with penalty-assisted," achieving a 19.3% improvement in overall performance score, an 18.3% increase in convergence speed, and a 15.3% rise in final reward value.

Sensitivity analysis revealed distinct mechanisms for reward and penalty components within the multi-objective reward function: reward components primarily drive system performance improvement and should be assigned higher weights; penalty components primarily serve as constraints and should maintain moderate weights to avoid excessive suppression of the system's exploration capabilities.

### 4.4. Simulation Setup

To simulate the uncertainties in real operational environments, a multi-level stochastic perturbation mechanism is designed. This study adopts a three-stage progressive blizzard intensity generation strategy, through dynamically adjusting parameters to simulate the real evolution of weather conditions. The domain of the blizzard intensity parameter is defined as [0.0, 0.8] and its assignment rule follows a piecewise linear regulation principle. In the initial stage of the training cycle, the blizzard intensity parameter is implemented with a linear increment strategy, starting from an initial value of 0.0 and increasing proportionally with training progress until reaching the upper threshold of 0.8; in the mid-training stage, the blizzard intensity parameter is maintained at the maximum threshold of 0.8 to simulate a prolonged heavy snowfall environment; in the late-training

stage, a random perturbation factor is introduced, causing the blizzard intensity parameter to follow a uniform distribution within the interval [0.0, 0.8] thereby enhancing the model's environmental adaptability.

Referring to the friction coefficient range of 0.25–0.35 for dry rails and wheels specified in the Railway Track Engineering Construction Quality Acceptance Standard, the midpoint value is adopted as the baseline for snow-free and ice-free conditions, defining the base friction coefficient as 0.3. As blizzard intensity increases and snow accumulation thickens, track surface roughness decreases, reducing the friction coefficient. A decay function is established by introducing a linear combination of blizzard intensity parameters and icing probability parameters. The spatiotemporal evolution of track occupancy status is modeled using a discrete-time Markov chain, with the state space defined as S = {Idle, Occupied}. The probability distribution satisfies the following: in the Idle state, the probability of remaining in the current state is 0.8, and the probability of transitioning to the Occupied state is 0.2; in the Occupied state, the probability of remaining in the current state is 0.7, and the probability of transitioning back to the Idle state is 0.3. Detailed parameter configurations are presented in Table 6.

**Table 6.** Simulation environment parameters for configuration.

| Parameter | Value Range | Default Value |
|---|---|---|
| Blizzard Intensity | [0.0, 0.8] | 0.2 |
| Icing Probability | [0.02, 0.5] | 0.02 |
| Base Friction Coefficient | [0.1, 0.3] | 0.3 |
| Track Occupancy Probability | [0.0, 1.0] | 0.2 |
| Maximum Delay Time | [0, 30] | 15 |

To quantify the impact of different blizzard intensities on high-speed rail operations, this study establishes a five-level blizzard intensity classification system and defines corresponding operational parameter adjustment criteria. As shown in Table 7, based on actual operational experience and meteorological data, blizzard intensity is categorized into five levels: no snow, light snow, moderate snow, heavy snow, and blizzard. Each level corresponds to specific blizzard intensity ranges, friction coefficient impacts, and maximum speed restrictions. This classification method not only considers the direct impact of blizzard intensity on train operation safety but also takes into account operational efficiency and practical feasibility, providing a quantitative basis for subsequent scheduling decisions.

**Table 7.** Weather levels and impact table.

| Blizzard Intensity Classification System | Blizzard Intensity Ranges | Friction Coefficient Impacts | Maximum Speed Restrictions |
|---|---|---|---|
| no snow | 0.0 | 100% | 350 km/h |
| light snow | (0.0, 0.3] | −20% | 300 km/h |
| moderate snow | (0.3, 0.6] | −40% | 250 km/h |
| heavy snow | (0.6, 0.8] | −60% | 200 km/h |
| blizzard | >0.8 | −80% | 150 km/h |

To comprehensively evaluate the effectiveness of the proposed scheduling method, this study selects two mainstream reinforcement learning algorithms—PPO and DQN—as baseline comparisons. Both algorithms are representative in the field of reinforcement learning and can reflect the difficulty of scheduling tasks and the adaptability of models from different perspectives. In terms of experimental design, all algorithms are trained and evaluated within the same high-speed rail scheduling simulation environment to

ensure a fair comparison. By comparing the performance of the three algorithms during the training process, their strengths and weaknesses in terms of final performance and stability are analyzed.

*4.5. Result Analysis*

To validate the superiority of the LSTM-PPO algorithm in high-speed rail dynamic scheduling under blizzard conditions, this study conducted comparative experiments, evaluating the proposed method against DQN and PPO algorithms. Figure 3 illustrates the training loss and average reward trends in the high-speed rail scheduling simulation environment. Left panel (Training Loss): The DQN algorithm exhibits a rapid decline in loss during the initial phase, stabilizing after approximately 400 training episodes with a minimum loss of ~0.99. While this indicates fast convergence, further improvements are limited, and minor fluctuations persist. The PPO algorithm maintains a relatively high loss level throughout training, demonstrating slower convergence but consistent stability with minimal fluctuations. In contrast, the LSTM-PPO algorithm initially shows higher loss values, which steadily decrease over training, ultimately achieving a final loss of 0.03—significantly outperforming both DQN and PPO. This highlights the enhanced capability of the LSTM-PPO framework to capture dynamic environmental features and optimize policy learning. Right panel (Average Reward): The DQN algorithm exhibits large initial reward fluctuations, dropping as low as −34.01, followed by gradual improvement and stabilization at ~9.11, reflecting moderate policy refinement. The PPO algorithm achieves a relatively high initial reward of ~11.43, maintaining stability throughout training, demonstrating robust baseline performance. The LSTM-PPO algorithm, however, demonstrates a continuous upward trend in average reward, ultimately reaching 21.37—substantially exceeding both DQN and PPO. This underscores the LSTM architecture's ability to enhance long-term reward accumulation and adaptability to complex temporal dependencies.

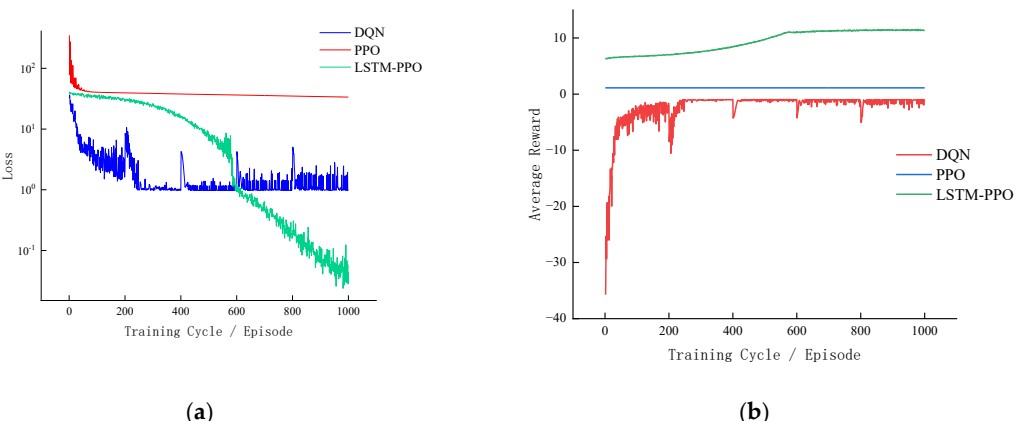

**Figure 3.** Comparative experimental results diagram. (**a**) Loss Function Comparison; (**b**) Reward Function Comparison.

This study demonstrates significant advantages in two core aspects: decision variable design and multi-objective handling. Traditional scheduling methods are constrained by discrete, single-dimensional decision variables and operate in a "memoryless" manner, failing to leverage historical state information. In contrast, the proposed agent's decision is represented as a multidimensional action vector, incorporating critical parameters such as speed restrictions and stop-time adjustments, enabling fine-grained control over train operations. The innovation lies in the LSTM-PPO model's integration of LSTM units to establish temporal dependencies in decision-making. The LSTM network dynamically encodes historical state sequences into a high-dimensional context vector, allowing deci-

sions to be based not on isolated current states but on a comprehensive understanding of the entire journey's dynamic evolution. This substantially enhances decision-making foresight and globality. As quantitatively validated in Figure 4, the LSTM-PPO's Value Loss curve achieves the lowest values and stabilizes around 20 in later training stages, whereas PPO's Value Loss stabilizes at ~40 (double the magnitude). This indicates that LSTM-PPO's history-informed decisions improve future cumulative reward estimation accuracy by nearly 50%, enabling superior long-term scheduling strategies. Regarding multi-objective handling, traditional methods rely on static weights predefined by expert experience, which struggle to adapt to dynamic environments. This study employs reinforcement learning algorithms to implicitly and adaptively balance multiple objectives. The integration of curriculum learning further enhances robustness in complex environments. By maximizing a composite cumulative reward, the model autonomously learns to trade-off sub-objectives across different states without manual intervention. Figure 4 experimentally validates these advantages: LSTM-PPO achieves a loss reduction to <30% of DQN/PPO levels within the first 10% of training, significantly shortening training time. Final losses are 60–90% lower than DQN/PPO in later stages, reflecting superior performance. The loss curves for LSTM-PPO exhibit smoother profiles with markedly smaller fluctuations compared to DQN/PPO, demonstrating enhanced stability and generalization. The LSTM architecture's temporal modeling capability—capturing historical information to improve policy and value estimation accuracy—proves particularly effective for tasks involving temporal dependencies.

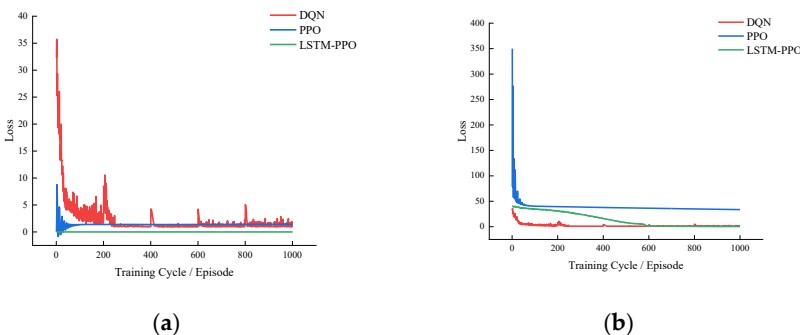

**Figure 4.** Comparative diagram of loss curves. (**a**) Policy Loss Comparison; (**b**) Value Loss Comparison.

Through reinforcement learning algorithms, the objective function evolves from traditional single-metric optimization to maximizing a comprehensive cumulative reward. As demonstrated in the final train schedule comparison (Figure 5), this integrated objective function yields significant results. After optimization by the three algorithms, cumulative train delays are substantially reduced. Departure time curves shift downward overall, with slopes markedly diminished. Comparisons reveal that relative to the unoptimized timetable, LSTM-PPO reduces delays to less than 5% of the original level, while PPO and DQN achieve reductions of approximately 25% and 16%, respectively. This optimization magnitude far exceeds that of the other two algorithms. This fully demonstrates that by maximizing a comprehensive reward function, the model can learn a scheduling strategy that is far superior to single-objective approaches, offering greater comprehensiveness and efficiency.

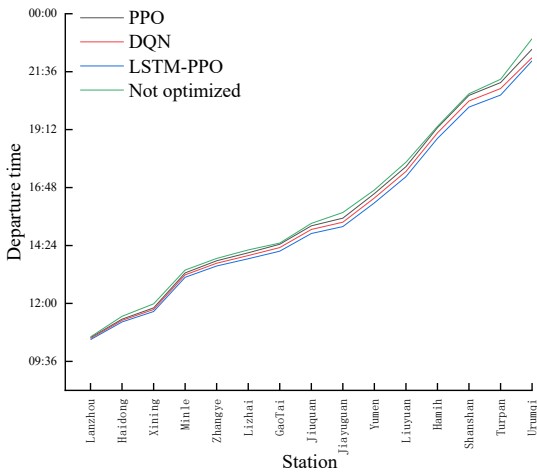

**Figure 5.** Comparative diagram of timetable results.

## 5. Conclusions

This study proposes an innovative solution for high-speed rail dynamic scheduling under blizzard conditions by integrating LSTM-based temporal perception with PPO reinforcement learning. A simulation experiment was designed using the Lanzhou-Xinjiang High-Speed Railway as the testbed. The results demonstrate the following:

(1) The LSTM module encodes spatiotemporal correlations of blizzard propagation and delay diffusion through hidden states, while the PPO module introduces a confidence-driven dynamic weight allocation mechanism to achieve real-time balancing between safety constraints and punctuality. This approach reduces strategy update variance by 62.5%, achieves over 95% delay compression, and improves average reward by 87.2%, significantly outperforming traditional DQN and PPO methods in temporal modeling, real-time responsiveness, and safety-efficiency trade-offs.

(2) A decoupled asynchronous update mechanism is developed, enabling phased training of the LSTM prediction module and PPO decision module to resolve gradient conflicts. A composite reward function incorporating six sub-objectives is designed with a dynamic weight adjustment mechanism. Through entropy regularization and curriculum learning strategies, the model autonomously optimizes objective priorities, reducing punctuality deviation by 96.8%.

(3) For actual railway operations, the deployment of the proposed method can be summarized into three practical steps: data integration, requiring real-time access to meteorological, train operation, and infrastructure monitoring data; system integration, embedding the scheduling optimization module into existing dispatching platforms to allow for interaction with human dispatchers; and operational implementation, promoting the method gradually through pilot testing on high-risk cold-region lines with compatibility checks against existing emergency protocols.

(4) Data availability and quality may hinder model accuracy in regions with incomplete monitoring infrastructure. Computational resources and real-time response remain a challenge for large-scale applications where high-dimensional optimization is required under strict time constraints. Furthermore, human–machine collaboration and safety certification demand transparent and explainable algorithm behavior, alongside redundancy mechanisms, to meet operational safety standards.

Currently, the focus is solely on a single route to simplify complex factors such as cross-line train interference and delay propagation coupling across multiple lines. This allows us to concentrate on core variables like friction coefficient decay during blizzard conditions and the three-stage evolution of blizzards, verifying their compatibility with

the LSTM-PPO algorithm. This ensures controllable experimental conditions and reproducible results. When subsequently expanding to multi-line network scheduling optimization, enhancing algorithm transparency and human–machine collaboration capabilities will be essential to meet operational safety standards. Crucially, collecting real-world operational and environmental data during actual extreme weather events will be indispensable for validating model assumptions, reducing uncertainty, and ensuring reliable practical application.

**Author Contributions:** Methodology, Y.L.; Writing—original draft, Z.C.; Writing—review & editing, N.W. All authors have read and agreed to the published version of the manuscript.

**Funding:** Natural Science Foundation of Gansu Province (25JRRA1852023-01-01).

**Data Availability Statement:** All data in this article was obtained from the National Railway Administration.

**Conflicts of Interest:** The author declares no conflicts of interest.

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
