# Peer review of "LSTM-PPO-Based Dynamic Scheduling Optimization for High-Speed Railways Under Blizzard Conditions"

_systems, doi:10.3390/systems13100884_

Round 1
Reviewer 1 Report
Comments and Suggestions for Authors
The paper has conducted relevant research on the problem of high-speed rail dynamic scheduling optimization and proposed the LSTM-PPO dual-module algorithm. The paper has high research significance and application value in the field of high-speed rail dynamic scheduling and extreme weather response. However, there are also the following issues in the whole manuscript that require further revision:
1)The language of the manuscript needs further improvement to enhance its readability; the authors are advised to organize the language throughout the manuscript and improve the overall writing quality.
2)It is suggested that the authors strengthen the description of research methods and innovation points in the abstract. Specific data should belong to research results, and the research conclusions should be summarized at the end of the abstract; the authors need to recondense and rewrite the article’s abstract.
3)In the section summarizing the research status, it is suggested that the authors condense the shortcomings of existing methods, so as to more intuitively demonstrate the research significance and value of the paper.
4)In conjunction with the third comment, it is suggested that the authors supplement the citation of relevant recent papers to reflect the latest research progress in the relevant research field.
5)The authors also need to carefully sort through the entire manuscript, as there are inconsistencies in the description of some key results. For example, in Table 6, when the snowstorm intensity is > 0.8, the friction coefficient only decreases by 20%, which contradicts the statement in the main text that "the friction coefficient decreases by 60%". This issue is crucial to the research of the entire manuscript, and the authors are expected to revise it carefully.
6)The authors should carefully revise the inconsistency in the expression of the parameter "Track Occupancy" (in Table 1) and "track occupancy ratio" (in the main text).
7)The meanings of the relevant symbols used in Equations (8) and (9) have not been explained.
8)The content of Figure 3(a) is identical to that of Figure 4(b). What is the authors' intention? What issue needs to be illustrated? The authors are requested to provide a corresponding explanation.
The paper has good research value and significance in the field of high-speed rail dynamic scheduling and extreme weather response. It is hoped that the authors will not limit themselves to the above-mentioned issues but carefully revise the entire manuscript.
Author Response
Comments 1:The language of the manuscript needs further improvement to enhance its readability; the authors are advised to organize the language throughout the manuscript and improve the overall writing quality.
Response 1: Thank you for pointing this out. We agree with this feedback. Accordingly, we have refined the content further, with the revisions located on pages 706-717.
Comments 2:It is suggested that the authors strengthen the description of research methods and innovation points in the abstract. Specific data should belong to research results, and the research conclusions should be summarized at the end of the abstract; the authors need to recondense and rewrite the article’s abstract.
Response 2: Agreed. Accordingly, we have revised the abstract to condense its content, emphasizing descriptions of the research methodology and innovative aspects. The modifications are located on pages 9–25.
Comments 3:In the section summarizing the research status, it is suggested that the authors condense the shortcomings of existing methods, so as to more intuitively demonstrate the research significance and value of the paper.
Response 3: Thank you for pointing this out. We agree with this comment. Accordingly, we have further refined the content, with the revisions located on pages 223-241.
Comments 4:In conjunction with the third comment, it is suggested that the authors supplement the citation of relevant recent papers to reflect the latest research progress in the relevant research field.
Response 4: Thank you for pointing this out. We agree with this comment. Accordingly, we have added two references, with the revisions located on page 153-157.
Comments 5:The authors also need to carefully sort through the entire manuscript, as there are inconsistencies in the description of some key results. For example, in Table 6, when the snowstorm intensity is > 0.8, the friction coefficient only decreases by 20%, which contradicts the statement in the main text that "the friction coefficient decreases by 60%". This issue is crucial to the research of the entire manuscript, and the authors are expected to revise it carefully.
Response 5: Agreed. Therefore, we have revised the unreasonable friction coefficient impact in Table 6. When snowstorm intensity > 0.8, the original friction coefficient is now modified to a 80% reduction. This table is located at page 637.
Comments 6:The authors should carefully revise the inconsistency in the expression of the parameter "Track Occupancy" (in Table 1) and "track occupancy ratio" (in the main text).
Response 6: Thank you for pointing this out. We agree with this comment. Therefore, we have changed “track occupancy rate” to “track occupancy ratio” to align with the terminology used in Table 1. The modification is located at page 291.
Comments 7:The meanings of the relevant symbols used in Equations (8) and (9) have not been explained.
Response 7: Agreed. We have added an explanatory note, this content is located at lines 348-364.
Comments 8:The content of Figure 3(a) is identical to that of Figure 4(b). What is the authors' intention? What issue needs to be illustrated? The authors are requested to provide a corresponding explanation.
Response 8: Thank you for pointing this out. We agree with this comment. We have revised Figure 3(a). The modification is located at line 670.

Reviewer 2 Report
Comments and Suggestions for Authors
This article proposes an LSTM-PPO framework for dynamic scheduling optimization of high-speed railways under blizzard conditions. It integrates temporal prediction with reinforcement learning and shows significant delay reduction in simulation.
Below are several points that, in my opinion, require revision or improvement:
1/ The paper lacks a clear description of how simulation scenarios were validated against real-world data a comparison with actual blizzard events would strengthen credibility.
2/ Model assumptions, e.g. friction reduction or icing probability, are not justified with reference. These parameters may critically affect results.
3/ The multi-objective reward function is well designed but very complex. Sensitivity analysis of weight selection would make results more robust.
4/ Many equations are difficult to read and could benefit from clearer formatting or a summary table of symbols and variables.
5/ The simulation setup focuses only on a single line. Adding tests on other networks or with different traffic patterns would improve generalization.
6/ The conclusion mostly restates results. It could better highlight practical implementation steps and limitations for deployment in real railway operations.
7/ The English is clear and well written, only minor sentence shortening could improve readability.
Author Response
Comments 1:The paper lacks a clear description of how simulation scenarios were validated against real-world data a comparison with actual blizzard events would strengthen credibility.
Response 1:Thank you for pointing this out. We agree with this observation. However, due to the difficulty in obtaining data from actual blizzard events in the real world, such studies have been conducted under hypothetical conditions (Li Dechang, Jia Ruogu, Li Wenhao, et al. Dynamic Scheduling Method for High-Speed Trains During Emergencies Caused by Strong Wind Natural Disasters [J]. Disaster Science, 2024, 39(03): 66-71.; Thaduri A, Garmabaki A, Kumar U. Impact of climate change on railway operation and maintenance in Sweden: A State-of-the-art review[J]. Maintenance, Reliability and Condition Monitoring (MRCM), 2021, 1(2): 52-70.;Veelenturf L P, Kidd M, Cacchiani V, et al. A railway timetable rescheduling approach for handling large-scale disruptions[J]. Erim Report, 2016, 50(3): 841-860.),we fully recognize that direct validation using real-world event data would significantly enhance the persuasiveness of this study. This limitation is now explicitly discussed in the conclusions of the revised manuscript, and we have identified data collection during future extreme events as a critical step for model refinement. The modifications are located on pages 751-761.
Comments 2:Model assumptions, e.g. friction reduction or icing probability, are not justified with reference. These parameters may critically affect results.
Response 2:Thank you for pointing this out. We agree with this observation. However, since the literature studying such standards does not explicitly address friction reduction or icing probability (Cui XQ, Zhou XL, Fu J, et al. Research on the Classification Standard for High-Impact Weather Conditions Affecting the Safe Operation of High-Speed Railways [J]. Journal of Disaster Science, 2016, 31(03): 26-30.; Leggett TS, Sdoutz G D. Liquid anti-icing chemicals on asphalt: friction trends[J]. Transportation research record, 2001, 1741(1): 104-113.; Ghaemi N, Cats O, Goverde R M P. Macroscopic multiple-station short-turning model in case of complete railway blockages[J]. Transportation Research Part C: Emerging Technologies, 2018, 89: 113-132. Therefore, referencing the friction coefficient range of 0.25-0.35 for dry rails and wheels specified in the Railway Track Engineering Construction Quality Acceptance Standard, the midpoint value is adopted as the baseline for snow-free and ice-free conditions. This is detailed in Section 4.4, with the revised content located on pages 613-619.
Comments 3:The multi-objective reward function is well designed but very complex. Sensitivity analysis of weight selection would make results more robust.
Response 3:Agreed. Therefore, we have supplemented the sensitivity analysis section by adding a new subsection 4.3 Sensitivity Analysis. The revised content is located on pages 570–598.
Comments 4:Many equations are difficult to read and could benefit from clearer formatting or a summary table of symbols and variables.
Response 4:Agreed. Therefore, we have consolidated all symbols and variables from the equations into the summary table. The modifications are located at page 270.
Comments 5:The simulation setup focuses only on a single line. Adding tests on other networks or with different traffic patterns would improve generalization.
Response 5:Thank you for pointing this out. We agree with this suggestion. Therefore, we have added clarification in the conclusion that we plan to expand to multi-line network scheduling optimization in the future. The revised content is located on pages 751-761.
Comments 6:The conclusion mostly restates results. It could better highlight practical implementation steps and limitations for deployment in real railway operations.
Response 6:Thank you for pointing this out. We agree with this comment. Therefore, we have added supplementary remarks to the conclusion section, with the revised content located on pages 738-750.
Comments 7:The English is clear and well written, only minor sentence shortening could improve readability.
Response 7:Thank you for pointing this out. We agree with this feedback. Accordingly, we have refined the content further, with the revisions located on pages 706-717.

Round 2
Reviewer 2 Report
Comments and Suggestions for Authors
Thank you for addressing my comments and improving the article. I have no further suggestions.